# Potential Benefits and Harms of Novel Antidiabetic Drugs during COVID-19 Crisis

**DOI:** 10.3390/ijerph17103664

**Published:** 2020-05-22

**Authors:** Maria Mirabelli, Eusebio Chiefari, Luigi Puccio, Daniela Patrizia Foti, Antonio Brunetti

**Affiliations:** 1Department of Health Sciences, University “Magna Graecia” of Catanzaro, Viale Europa, 88100 Catanzaro, Italy; maria.mirabelli@unicz.it (M.M.); echiefari@gmail.com (E.C.); foti@unicz.it (D.P.F.); 2Complex Operative Unit of Endocrinology and Diabetes, Hospital Pugliese-Ciaccio, 88100 Catanzaro, Italy; puccio55@libero.it

**Keywords:** COVID-19, diabetes, DPP4 inhibitors, GLP1 receptor agonists, SGLT2 inhibitors

## Abstract

Patients with diabetes have been reported to have enhanced susceptibility to severe or fatal COVID-19 infections, including a high risk of being admitted to intensive care units with respiratory failure and septic complications. Given the global prevalence of diabetes, affecting over 450 million people worldwide and still on the rise, the emerging COVID-19 crisis poses a serious threat to an extremely large vulnerable population. However, the broad heterogeneity and complexity of this dysmetabolic condition, with reference to etiologic mechanisms, degree of glycemic derangement and comorbid associations, along with the extensive sexual dimorphism in immune responses, can hamper any patient generalization. Even more relevant, and irrespective of glucose-lowering activities, DPP4 inhibitors and GLP1 receptor agonists may have a favorable impact on the modulation of viral entry and overproduction of inflammatory cytokines during COVID-19 infection, although current evidence is limited and not univocal. Conversely, SGLT2 inhibitors may increase the likelihood of COVID-19-related ketoacidosis decompensation among patients with severe insulin deficiency. Mindful of their widespread popularity in the management of diabetes, addressing potential benefits and harms of novel antidiabetic drugs to clinical prognosis at the time of a COVID-19 pandemic deserves careful consideration.

## 1. Introduction

The clinical spectrum of Coronavirus Disease 2019 (COVID-19), caused by a novel coronavirus named Severe Acute Respiratory Syndrome Coronavirus 2 (SARS-CoV-2), ranges from asymptomatic presentation or mild upper respiratory symptoms to severe pneumonia with respiratory failure, acute respiratory distress syndrome (ARDS) and death due to sepsis and septic shock [1]. Although the most commonly reported are flu-like symptoms, with fever, fatigue, headache, sore throat, coughing and dyspnea, a gastrointestinal symptomatology can be frequently encountered in clinical practice, representing the COVID-19 patient’s chief complaint in some cases [2]. In addition, progressive and diffuse vascular derangement represents a major threat in severe COVID-19 presentation, with histological evidence of endothelial inflammation, congestion and thrombosis in several internal organs, including the heart, kidneys, lungs, liver and intestines [3]. Postmortem studies have underlined luminal and mural fibrin deposition in pulmonary septal capillaries and a disseminated microvascular injury, reminiscent of other thrombotic microangiopathies, but with distinct features [4,5]. In addition, consistent with an inflammation-induced procoagulant shift of endothelial cells, the clinical course of COVID-19 can be occasionally worsened by deep vein thrombosis and pulmonary embolism [5]. Based on current available evidence, patients with diabetes are expected to have enhanced disease severity and an increased risk of being admitted to the intensive care unit (ICU) with respiratory failure and multiorgan dysfunction following SARS-CoV-2 infection [1,6]. When compared with patients who did not receive ICU care, a larger fraction of ICU patients with COVID-19 were found to have underlying diabetes (22.2% vs. 5.9%), presumably type 2 (T2D) [1]. These preliminary observations from Hubei, the Chinese province where the outbreak began, have also been recently corroborated by the Lombardy Intensive Care Unit (ICU) Network, which found T2D as the fourth most common comorbidity among 1591 critically ill patients admitted to the ICU with COVID-19 (180 patients, 17%) after the spread of the epidemic in Northern Italy and across Europe [7]. Even more relevant, one-third of COVID-19 patients dying in Italy have T2D [8], congruous with the independent association of this condition with fatal complications during two other coronavirus-related respiratory infection epidemics, such as the Severe Acute Respiratory Syndrome (SARS) in 2002, and the Middle East Respiratory Syndrome (MERS) in 2012 [1].

Given the extremely high global prevalence of diabetes, affecting more than 450 million adults worldwide and still on the rise, in accordance with the last International Diabetes Federation (IDF) atlas [9], the COVID-19 pandemic represents a serious threat to a large vulnerable population, so that intensive supportive care, plus optimum pharmacological management, are critical to address the clinical course of COVID-19 and prevent fatal outcomes. Herein, we provide a brief narrative overview of mechanisms, potential benefits and safety issues of novel classes of antidiabetic drugs during the COVID-19 crisis, with a focus on infectious outcomes.

## 2. DPP4 Inhibitors

Dipeptidyl-peptidase 4 (DPP4) inhibitors, also known as gliptins, represent a relatively new class of oral antidiabetic agents that block the inactivation of the distal gut-derived insulinotropic hormone glucagon-like peptide 1 (GLP1). By positively affecting glucose control with minimal risk of hypoglycemia, these drugs have gained popularity as a second-line option for managing T2D due to their favorable side effect profile and competitive costs [10]. To date, there are five DPP4 inhibitors available on the European market, including sitagliptin, saxagliptin, linagliptin, alogliptin and vildagliptin, all of which have specific pharmacodynamic and pharmacokinetic properties, with potentially relevant implications for patients affected by liver or kidney disease, notwithstanding comparable glycemic efficacy and inhibition of DPP4 activity [11]. According to mechanistic studies, DPP4, formerly known as T-cell antigen CD26, is a multifunctional soluble and cell-bound serine protease, abundantly expressed in lymphocytes and adipocytes as well as in many other cell types, including endothelial and epithelial cells, which plays critical roles in the modulation of glucose homeostasis and inflammatory responses [10]. Interestingly, DPP4 was identified as a functional receptor for human coronavirus-Erasmus Medical Center (hCoV-EMC) [12], the virus responsible for MERS and genetically close to SARS-CoV-2. In addition, antibodies directed against DPP4 could impair hCoV-EMC infection in primary human bronchial epithelial cells [12]. The observation that the S1 domain of SARS-CoV-2 spike glycoprotein interacts with the host receptor protein DPP4 [13,14] raised the hypothesis that SARS-CoV-2 may use DPP4 as a (further) functional receptor to gain entry into the host, in addition to the well-documented angiotensin-converting enzyme 2 (ACE2) [15]. Although a direct involvement of DPP4 in the modulation of SARS-CoV-2 entry has been denied in a stably transfected human embryonic 293T cell line [16], this computational evidence [13] has raised speculation and questions about whether targeting one of the potential host determinants of virulence with DPP4 inhibitors would be useful in attenuating COVID-19 infection after viral exposure [17,18], especially in patients with T2D, typically depicted by a pathogenic dysregulation of DPP4 expression [19]. Apart from the anti-hyperglycemic effect related to protease-dependent modulation of the entero-insular axis, DPP4 inhibitors possess unique immunomodulatory therapeutic potential for autoimmune and rheumatological disorders [20,21], cancer [22] and, most remarkably, MERS [23]. Targeting DPP4 to inflect natural viral dynamics has been thus suggested as a pharmacologically reasonable strategy in the case of COVID-19, as well as other severe respiratory diseases related to coronaviruses [17,23]. Interestingly, rheumatological disorders and severe COVID-19 share the overproduction of NF-kB-dependent proinflammatory cytokines and mediators, such as IL-1, IL-6 and TNF-α, as a common key pathogenetic mechanism [24,25]. The blockade of IL-6 receptors with tocilizumab, a humanized IgG1 monoclonal antibody commonly used in patients with rheumatoid arthritis, is currently under investigation in patients with severe COVID-19 pneumonia, whereas inhibition of IL-1 release by hydroxychloroquine, a first-line disease-modifying antirheumatic drug (DMARD) therapy, is under investigation in patients with mild to moderate COVID-19 to prevent disease progression (www.aifa.gov.it). In addition to its immunomodulant and anti-inflammatory effects, hydroxychloroquine can interfere with the glycosylation of SARS-CoV-2 spike proteins and ACE2 receptors, blocking, in in vitro experimental models, viral entry [26], so that its empiric prophylactic use among patients and healthcare workers at high risk of infection has been recently advocated by the Indian Council of Medical Research (https://icmr.nic.in/content/covid-19). It is also noteworthy that, during infection progression, SARS-CoV-2 decreases the activity and expression of ACE2 [27], the primary component of an alternate renin-angiotensin system (RAS) that counteracts angiotensin II (ANGII). The unbalanced ANGII activity has been indicated as a critical driver for severe COVID-19 manifestations in lungs and other organs [28]. In this regard, DPP4 inhibitors could interfere with the RAS system and reduce ANGII levels, thereby ameliorating hypertension and comorbid cardiac remodeling in experimental animal models [29]. However, despite a plausible pharmacological rationale, validation of drugs with originally different therapeutic indications for preventing or treating COVID-19 can be challenging even in the case of DPP4 inhibitors, as current computational or preclinical evidence does not suffice. To date (May 2020), neither data are available regarding clinical and demographic characteristics of people with diabetes developing severe COVID-19 complications, nor do we know whether background glycemic levels or antidiabetic therapies might have a role. Epidemiological and experimental clinical research programs are comprehensibly more difficult during a pandemic because of many possible barriers, including risk of infection for healthcare workers and research staff deployment to provide clinical care [30]. A metanalysis evidenced that treatment with DPP4 inhibitors does not increase the overall risk of respiratory infections with respect to placebo or conventional oral antidiabetic comparators (metformin, sulfonylureas, thiazolidinediones; OR 0.98, 95% CI 0.91 to 1.05), although no subgroup analysis has been performed concerning relationships with specific viral pathogens [31]. More recently, a large prospective cohort study in France evidenced an inverse association between diabetes and non-influenza respiratory virus illnesses over the course of three flu seasons [32]. The study ended before the COVID-19 pandemic; however, hospitalized patients with flu-like symptoms who tested positive for non-influenza respiratory viruses, including coronavirus 229E, were less likely to have diabetes than patients infected with seasonal influenza viruses (18% vs. 25%). Unfortunately, background information on medication use was not assessed in this study, except for immunosuppressive drugs [32]; therefore, whether specific antidiabetic medications would have prevented non-influenza respiratory virus-related hospitalization in patients with diabetes remains uncertain. Interestingly, and regardless of the viral strain, coronavirus infections are counteracted by submucosal mast cells of the respiratory tract as the forefront part of the innate immunity response [33]. Endothelial transmigration of mast cells is under the control of the stromal derived factor 1 (SDF-1), which is cleaved and inactivated by DPP4 [34,35]. Because of its dose-dependent ability to selectively induce the production of IL-8, but not other pro-inflammatory mediators from mast cells, SDF-1 orchestrates leukocyte recruitment and vascular remodeling in the context of arteriogenesis, both of which could be boosted by DPP4 blockage in experimental animal models [34,36]. In addition, increased plasmatic levels of SDF-1 have been associated with maturation and migration of leukocytes to mucosal compartments and immune response-dependent abortion of human immunodeficiency virus 1 (HIV1) infection [37]. An intriguing question that could be raised is whether DPP4 inhibitor-related augmentation of SDF-1 levels and mast cells in mucosal entry sites would confer resistance against SARS-CoV-2 and other coronaviruses. In this regard, some evidence suggests that, while neutralizing these pathogens, mast cells release antiviral and cytotoxic mediators which, in some circumstances, can aggravate the clinical course of the disease [33]. The dual effects of mast cells in viral infections could introduce further arguments and controversies in the debate about the use of DPP4 inhibitors as a potential protective strategy against severe COVID-19 manifestations [38]. While downregulation and inhibition of DPP4 activity by MERS and HIV1 have been demonstrated [38], to date, there is no indication that binding of SARS-CoV-2 would lead to similar effects. In addition, concerning COVID-19 related vascular damage, it should be noted that, in contrast to preclinical evidence, human studies have shown either a neutral [39] or a detrimental [40] effect of DPP4 blockage on endothelial function in the setting of diabetes. Figure 1 summarizes the potential benefits and harms of DPP4 inhibitors on COVID-19 outcomes.

## 3. SGLT2 Inhibitors

Sodium-glucose cotransporter 2 (SGLT2) inhibitors, also known as gliflozins, are the last class of antidiabetic drugs approved by the US Food and Drugs and European regulatory agencies (FDA and EMA, respectively) for the treatment of T2D [41]. Four SGLT2 inhibitors are currently available on the European market, namely canagliflozin, dapagliflozin, empagliflozin and ertugliflozin, with comparable glycemic efficacy despite differences in terms of selectivity toward SGLT2 [42]. Concerning their pharmacological mechanism of action, SGLT2 inhibitors suppress renal glucose reabsorption and enhance urinary glucose excretion in an insulin-independent manner, being therefore suitable for patients with a marked decline in β-cell function because of autoimmune and inflammatory processes, including pancreatitis, cystic fibrosis and hemochromatosis, thereby allowing off-label prescriptions in common practice [41,43]. Clinical trials and post-marketing surveillance have drawn attention to the increased risk of genitourinary infections and volume depletion accompanying the use of SGLT2 inhibitors in the presence of diabetes [41]. Yet, it is of the utmost significance, in the time of this COVID-19 pandemic, the exceptional risk of diabetic ketoacidosis (DKA) associated with SGLT2 inhibitors among vulnerable patient categories, such as long-standing T2D patients with marked β-cell insufficiency or patients with autoimmune diabetes, during an intercurrent illness [44]. DKA, a potentially life-threatening metabolic complication of diabetes, may occur as a consequence of absolute or relative insulin deficiency, and is typically characterized by hyperglycemia (>250 mg/dL, generally 350–800 mg/dL), ketosis (plasma β-hydroxybutyrate 4.2–11.0 mmol/L) and acidosis (pH < 7.3) [45]. When related to SGLT2 inhibition, DKA may be even more deceitful with an “euglycemic” presentation (<250 mg/dL) because of an increased urinary loss of glucose that mitigates hyperglycemia [45]. Very recently, the FDA raised a warning about real-world off-label use of SGLT2 inhibitors among patients with autoimmune diabetes, given its association with a DKA incidence rate of 7.3/100 person-years, 2.6-fold higher than what would be expected based on clinical trial data, with a peak of about 19.7/100 person-years in young adult women (22–44 years of age) [44]. The extremely high DKA incidence in young women with autoimmune diabetes may be related to distorted eating attitudes and behaviors, anxiety about body weight and size and lower adherence toward insulin replacement therapy, frequently observed in this particular population [46]. However, it should be noted that this sexual dimorphism also exists for the genitourinary side effects of SGLT2 inhibitors [41], which represent potential precipitating factors for DKA, along with respiratory and gastrointestinal infections such as COVID-19 [47]. Severe COVID-19 outcomes, including lethality, appear less frequent in women [1,7], probably due to sex-based immunological responses [48], although discrepancies in smoking habits [49], the prevalence of hypertension and specific multimorbidity combinations [50], in contrast with the intrinsic vulnerability of pregnancy status [51], might be no less important. In light of this, whether the female gender would still have a favorable impact on severe COVID-19 outcomes in the presence of diabetes remains to be defined, so that a gender bias could be avoided when providing medical care. COVID-19 may increase insulin demand and induce fever, nausea and anorexia with consequent hyperketonemia, which accentuates the gastrointestinal symptoms of infection in a vicious cycle. Metabolic decompensation toward DKA, either hyperglycemic or “euglycemic”, in susceptible diabetic patients on SGLT2 inhibitors can be further exacerbated by volume depletion from persistent glycosuria. At initial symptoms of COVID-19 illness, patients with off-label prescription or long-lasting T2D with severe β-cell insufficiency requiring insulin therapy should temporarily stop the SGLT2 inhibitor, contact their medical provider, monitor capillary blood ketones and take supplemental boluses of rapid insulin along with liquids and carbohydrates. A full episode of COVID-19-related DKA can be successfully prevented with these measures. Nevertheless, it should be recognized that the pharmacological effects of SGLT2 inhibitor treatment may persist beyond several half-lives of elimination, and some patients will still require hospitalization for DKA even after stopping the SGLT2 inhibitor, with glycosuria and ketonemia persisting for up to 9–10 days [52].

## 4. GLP1 Receptor Agonists

GLP1 receptor agonists (GLP1 RAs), also known as incretin-mimetics, provide pharmacologic levels of exogenous GLP1 which, analogously to the endogenous gut-derived incretin hormones, improve glucose homeostasis through enhanced glucose-dependent insulin secretion, delayed gastric emptying and increased post-meal satiety [53,54]. By providing an additional benefit of weight loss, GLP1 RAs represent the preferable second-line option for patients with obesity and inadequately controlled T2D, as adjunct to lifestyle interventions and metformin [55]. In addition, due to their claimed protective effects in cardiovascular outcome clinical trials, GLP1 RAs and SGLT2 inhibitors have been recently endorsed by the American Diabetes Association as the most appropriate second-line treatment for T2D patients with established atherosclerotic cardiovascular disease (ASCVD) or a high related risk (≥55 years of age with coronary, carotid or lower-extremity artery stenosis >50% or left ventricular hypertrophy), irrespective of glycemic levels [56]. To date, several GLP1 RAs with different half-lives and pharmacokinetics properties have been licensed in Europe for subcutaneous administration: twice-daily exenatide; once-daily liraglutide and lixisenatide and once-weekly dulaglutide, albiglutide and semaglutide [57], whereas the once daily preparation of semaglutide for oral administration is forthcoming (www.ema.europa.eu/). Similar to DPP4 inhibitors, systemic anti-inflammatory effects have been associated with the use of GLP1 RAs in patients with T2D, as a consequence of the inhibitory activity on cytokine release, due to their interference with NF-kB signaling pathways [58]. In animal models of sepsis, administration of liraglutide has been proved to improve survival and vascular dysfunction, along with inflammatory and hemostatic parameters [59]. In addition, given the contribution of GLP1 receptors in airway remodeling [60], several clinical trials are underway to address the theoretical potential of GLP1 RAs for the amelioration of chronic obstructive respiratory diseases. Even more relevant, in animal studies, the GLP1 RA liraglutide has been associated with the upregulation of ACE2, a cell-bound protease, abundantly expressed in alveolar epithelial cells, enterocytes and vessels upstream of the counter-regulatory RAS pathway, which exerts a negative effect on inflammatory and fibrotic processes [61]. Although still unsupported by clear translational evidence, the GLP1 RAs-induced upregulation of ACE2 could ameliorate lung injury during COVID-19, antagonizing the reduction of ACE2 expression levels that are hallmarks of infection progression [15,27] and preventing the over-activated immune response critical for ARDS [62]. On the other hand, ACE2 enables virus entry into host target cells [14,15,26], raising speculation of increased susceptibility to COVID-19 infection in the case of ACE2 overregulation, as a consequence of long-term treatment with ACE inhibitors and/or ANGII receptor blockers, GLP1 RAs or a combination of both in hypertensive patients with diabetes [15]. Ex vivo and in vitro demonstrations of viral particle aggregates within human kidney endothelial cells [3] and engineered blood vessel organoids [63] suggest that the SARS-CoV-2 virus could target the endothelium via the locally expressed surface receptor ACE2, triggering a systemic vascular dysfunction that shifts vascular balance towards inflammation and hypercoagulability, leading to multiorgan failure in most critical cases [3,64]. However, to date, no robust clinical-epidemiological studies have been put forward concerning the correlation between these medications use and COVID-19 severity, adjusting for potential confounding variables such as sex, age and comorbidities possibly affecting ACE2 expression patterns. The issue of whether treatment-induced overregulation of ACE2 in target tissues would lead to an increased risk of SARS-CoV-2 infection, or its pulmonary-protective activity would prevail, has yet to be solved. Hence, current position statements of the Council of Hypertension of the European Society of Cardiology (www.escardio.org) clarify that ACE inhibitors and angiotensin II receptor blockers should be continued and prescribed according to previously established guidelines. At the same time, despite experts’ suggestions to reconsider diabetes management on a case-by-case basis, in view of patient setting and clinical judgment [65,66], no specific recommendations on the initiation and use of GLP1 RAs during the COVID-19 pandemic have been provided to diabetes specialists. On par with the abovementioned antihypertensive drugs, we believe that there is no reason supported by evidence to contraindicate, or vice versa, overpromote GLP1 RAs for managing people with T2D; however, there is an urgent need to determine whether their use can influence the clinical course of COVID-19. In this regard, as it has been remarked [66], the current evidence with GLP1 RAs and DPP4 inhibitors in critically ill diabetic patients with COVID-19 complications is inadequate to make therapeutic recommendations, and insulin therapy should be safely sustained as the preferred option for managing diabetes. On a different note, we retain that special consideration should be also paid to the eventuality of gastrointestinal symptoms related to GLP1 RAs when patients recover from COVID-19. Nausea, vomiting and diarrhea have been reported in up to 30%, 15% and 15%, respectively, of diabetic patients on GLP1 RAs [67]. Usually, these gastrointestinal disorders arise during the first few weeks of treatment and are more pronounced on a background of metformin or insulin therapy [67]. However, while metformin can directly elicit a mild gastrointestinal intolerance, the correlation with insulin therapy may underline a long-standing T2D with poor glycemic control, a condition commonly associated with impaired gastric motility, which may accentuate the deceleration of gastric emptying induced by exogenous GLP1 RAs [67]. Very recently, a meta-analysis of 60 studies, including 4243 patients with COVID-19, has evidenced that anorexia, nausea, vomiting, diarrhea, abdominal pain or discomfort are common manifestations of COVID-19, especially in severe forms of infection requiring hospitalization and ICU care (17.1%) [2]. Notwithstanding a postulated prognostic relevance, gastrointestinal symptoms may also accompany the course of milder forms of COVID-19 [2,68], and appear associated with prolonged illness [68]. Given the sustained attenuation in appetite sensations and dietary energy intake, coupled with gastrointestinal intolerance, the use of GLP1 RAs may further delay patient recovery and a temporary suspension should be cautiously considered in the most fragile cases. Figure 2 summarizes the potential benefits and harms of GLP1 RAs on COVID-19 outcomes.

## 5. Optimizing Glycemic Control

According to a recent international panel of experts [69], people with diabetes who have not yet been infected with COVID-19 should intensify their glycemic control as a means of primary prevention for a severe illness. It is a well-known fact that poorly controlled hyperglycemia impairs the immune functions and worsens the clinical course of infections in all forms of diabetes [70,71]. In addition, with reference to the particular case of COVID-19, it has been evidenced that glycosylation of ACE2 receptors, a post-translational reaction that can be boosted by hyperglycemia, is necessary for cell entry, as ACE2 alone does not suffice [72]. Lowering the amount of glycosylated ACE2 with anti-hyperglycemic measures in diabetic patients may thus reduce the burden of SARS-CoV-2 infection in case of viral exposure. To optimize diabetes management, previously established international guidelines [55,56], with relevance to specific national adaptations and reimbursement criteria for glucose-lowering drugs, must be considered. However, when patients with diabetes develop COVID-19, treatment with SGLT2 inhibitors or GLP1 RAs should deserve careful attention on a case-by-case basis, dependent on their own clinical presentation and concurrent comorbidities. When these types of treatments need to be discontinued due to COVID-19, the alternative choice is insulin, irrespective of the patient setting [66,70]. Table 1 reassumes the efficacy and safety concerns around novel antidiabetic drugs, with special reference to COVID-19 outcomes.

## 6. Conclusions

Even if current practices for treating diabetic patients infected by SARS-CoV-2 are no different from those of the general population, special care is required for patients with diabetes, given the greater risk of complications until in-hospital death. Furthermore, some novel antidiabetic drugs, such as SGLT2 inhibitors and GLP1 RAs, routinely employed in T2D management, may exacerbate the clinical expression of COVID-19 and, thus, should be stopped in the most fragile patients. Conversely, although not univocal, some evidence suggests that DPP4 inhibitors may retain the potential for preventing and/or mitigating the clinical course of COVID-19. Considering the minimal risk upon use, the relatively long experience and their easy availability worldwide, DPP4 inhibitors might be taken into account for clinical use as experimental drugs. However, in order to justify either the overpromotion or the off-label use for this drug class, reliable and unbiased observational research, addressing clinical outcomes and predictors in the case of SARS-CoV-2 exposure, is a fundamental prerequisite, as no patient generalization can be applied in the setting of diabetes.

## Figures and Tables

**Figure 1 ijerph-17-03664-f001:**
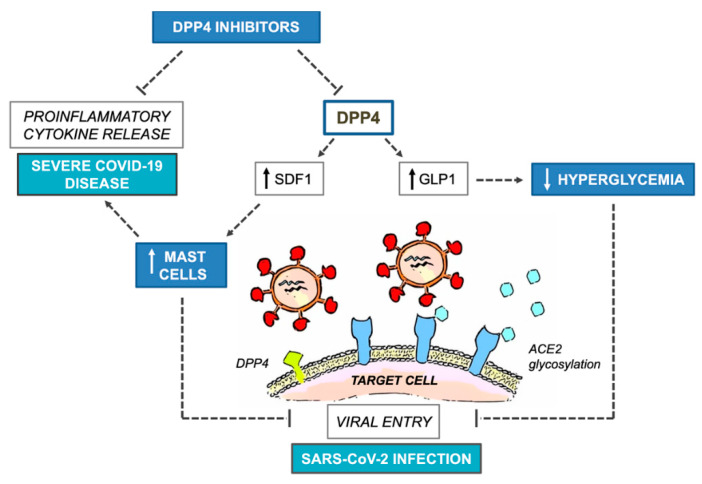
Potential benefits and harms of DPP4 inhibitors on COVID-19 outcomes. COVID-19, Coronavirus Disease 2019; DPP4, dipeptidyl-peptidase 4; SDF1, stromal derived factor 1; GLP1, glucagon-like peptide 1; ACE2, angiotensin converting enzyme 2; SARS-CoV-2, Severe Acute Respiratory Syndrome Coronavirus 2.

**Figure 2 ijerph-17-03664-f002:**
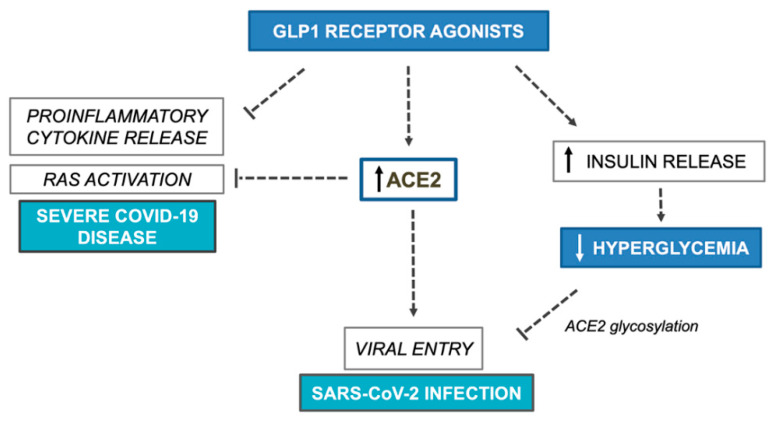
Potential benefits and harms of GLP1 receptor agonists on COVID-19 outcomes. GLP1, glucagon-like peptide 1; RAS, renin angiotensin system; ACE2, angiotensin converting enzyme 2; COVID-19, Coronavirus Disease 2019; SARS-CoV-2, Severe Acute Respiratory Syndrome Coronavirus 2.

**Table 1 ijerph-17-03664-t001:** Summary of efficacy and safety concerns of novel antidiabetic drug classes.

Drug Class	Efficacy	Safety Issues	Ref.	Comments on COVID-19
DPP4 inhibitors	Moderate glucose-lowering effect Neutral effect on body weight Low risk of hypoglycemia	FDA warning over acute pancreatitis risk	[73]	Potential reduction of COVID-19 infection severity
SGLT2 inhibitors	Moderate glucose-lowering effect Moderate weight loss Low risk of hypoglycemia	FDA warning over DKA risk High risk for genitourinary infections Contraindications for eGFR < 45mL/min	[42]	Monitoring of capillary blood ketones plus adequate hydration and carbohydrate intake at first symptoms of COVID-19 infection Consider transitory suspension for patients at high risk for DKA decompensation
GLP1 receptor agonists	Elevate glucose-lowering effect Moderate weight loss Low risk of hypoglycemia	Loss of appetite, nausea, diarrhea FDA warning over acute pancreatitis risk	[57]	Controversy regarding susceptibility and severity of COVID-19 infection Consider transitory suspension for patients with prolonged COVID-19 gastrointestinal symptomatology

COVID-19, Coronavirus Disease 2019; DPP4, dipeptidyl-peptidase 4; SGLT2, sodium-glucose cotransporter 2; FDA, US Food and Drugs Administration; DKA, diabetic ketoacidosis; eGFR, estimated glomerular filtration rate; GLP1, glucagon-like peptide 1.

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
