# Peer review of "Potential Benefits and Harms of Novel Antidiabetic Drugs During COVID-19 Crisis"

_ijerph, 2020, doi:10.3390/ijerph17103664_

Round 1
Reviewer 1 Report
Maria Mirabelli et al., give a short but detailed overview of the potential benefits and harms of antidiabetic drugs in patients suffering from diabetes and COVID-19. In principal the paper is written very well and it is a very interesting up-to-date topic. However, in my opinion there are several points that need to be carefully addressed and clarified.
SARS-CoV-2 recognizes ACE2 as its receptor. MERS-CoV recognizes DPP4 (CD26) as its receptor. The authors should inform the reader whether SARS-CoV-2 also recognizes DPP4 as a receptor and whether SARS-CoV-2 can enter a cell via DPP4.
Please explain in detail why targeting DPP4 has been suggested as pharmacologically reasonable strategy to treat COVID-19 patients (line 77-79).
Severe COVID-19 is described to be associated with damage of small vessels, endothelial damage, thrombus and edema formation (more or less in the whole body and not only in the lung and heart). Please supply some detailed information on that in the introduction.
Besides other cells, DPP4 is expressed in endothelial cells. Might an interaction of SARS-CoV-2 spike protein with endothelial cells might be causative for the endothelial damage? Might DPP4 inhibitor protect or result in (increased) endothelial damage? DPP4 inhibitors have been discussed to promote collateral artery growth in diabetes and accordingly promote vascular cell proliferation (Vedantham et al., Cells 2018, 7, 181; doi:10.3390/cells7100181). So the question arises whether DPP4 inhibitors might show positive or negative effects on vascular cells and or vascular cell proliferation.
Does the interaction between the SARS-CoV-2 spike protein and DPP4 result in activation or inactivation of the enzyme DPP4?
I think this is an important point since blocking DPP4 interferes with SDF-1 alpha cleavages and accordingly results in increased levels of SDF-1 alpha (an information which should also be supplied to the reader). The receptor for SDF-1 is CXCR4, which is expressed on leukocytes and mast cells. Increased levels of SDF-1 result in increased levels of leukocytes and mast cells, and mast cells in turn recruit even more leukocytes. Although it is likely that increased levels of SDF-1 due to DPP4 inhibition show positive effects in diabetic patients (for detailed information please see S Vedantham et al., Cells 2018, 7, 181; doi:10.3390/cells7100181), it is not clear what might happen in diabetic patients infected with SARS-CoV-2. Although SDF-1 mediated increased levels of leukocytes may show positive effects in the early phase of COVID-19 in terms of combating viral infection, the question is what happens when mast cells get recruited and activated (what is very likely in SARS-CoV-2 infected patients) and in turn recruit even more leukocytes? Does this result in adverse effects such as sustained severe inflammation as observed in COVID-19 patients? In that context, the role of mast cells in COVID-19 should be discussed in detail and the authors should comment on the positive or may be even negative effects of mast cells in COVID-19.
Moreover, I think it would be very helpful for the reader to show a sketch explaining the mechanisms potentially resulting in positive or negative effects (if existing) of DPP4 inhibitors in diabetic patients with SARS-CoV-2 infection.
I personally would be very happy of DPP4 inhibitors would show beneficial effects in COVID-19 patients, but for me this is not really obvious after reading this short article. However, this review article offers an excellent opportunity to discuss the potential effects of DPP4 inhibitors in detail and either support or contradict the use this drug class in COVID-19.
Together, I think it needs to be very critically addressed whether DPP4 inhibitors may show beneficial or harmful effects in diabetic (and may be also non-diabetic) individuals infected with SARS-CoV-2. The Abstract and the conclusion should be modified accordingly.
GLP1 receptor agonists: The paper would again profit from a figure illustration the effect of GLP1 receptor agonists on upregulation of ACE2 along with the potential positive and negative effects in corona infected patients.
Table 1: Please supply references for statements on efficacy and safety issues for each drug mentioned.
Author Response
Authors: We thank the reviewer for the careful reading of the manuscript. We have accepted most of the comments raised. Please find below a detailed point-by-point response (reviewers' comments in black, our replies in blue italic).
Maria Mirabelli et al., give a short but detailed overview of the potential benefits and harms of antidiabetic drugs in patients suffering from diabetes and COVID-19. In principal the paper is written very well and it is a very interesting up-to-date topic. However, in my opinion there are several points that need to be carefully addressed and clarified.
SARS-CoV-2 recognizes ACE2 as its receptor. MERS-CoV recognizes DPP4 (CD26) as its receptor. The authors should inform the reader whether SARS-CoV-2 also recognizes DPP4 as a receptor and whether SARS-CoV-2 can enter a cell via DPP4.
Authors: In contrast to the well-documented functional role of ACE2 in the modulation of SARS-CoV-2 entry into the host, the significance of DPP4 receptor is still under investigation. We now refer to this point (lines73-81).
Please explain in detail why targeting DPP4 has been suggested as pharmacologically reasonable strategy to treat COVID-19 patients (line 77-79).
Authors: In addition to the anti-inflammatory effects, DPP4 inhibitors retain the potential to modulate the renin angiotensin system (RAS) that is critical for severe manifestations of COVID-19 in lungs and other organs (this is now detailed in lines 102-108).
Severe COVID-19 is described to be associated with damage of small vessels, endothelial damage, thrombus and edema formation (more or less in the whole body and not only in the lung and heart). Please supply some detailed information on that in the introduction.
Authors: We now refer to this aspect of COVID-19 (lines 37-39 and 238-242).
Besides other cells, DPP4 is expressed in endothelial cells. Might an interaction of SARS-CoV-2 spike protein with endothelial cells might be causative for the endothelial damage? Might DPP4 inhibitor protect or result in (increased) endothelial damage? DPP4 inhibitors have been discussed to promote collateral artery growth in diabetes and accordingly promote vascular cell proliferation (Vedantham et al., Cells 2018, 7, 181; doi:10.3390/cells7100181). So the question arises whether DPP4 inhibitors might show positive or negative effects on vascular cells and or vascular cell proliferation.
Authors: We now discuss the potential role of DPP4 inhibitors on vascular remodeling and endothelial function ( lines 131-135 and 144-148).
Does the interaction between the SARS-CoV-2 spike protein and DPP4 result in activation or inactivation of the enzyme DPP4? I think this is an important point since blocking DPP4 interferes with SDF-1 alpha cleavages and accordingly results in increased levels of SDF-1 alpha (an information which should also be supplied to the reader). The receptor for SDF-1 is CXCR4, which is expressed on leukocytes and mast cells. Increased levels of SDF-1 result in increased levels of leukocytes and mast cells, and mast cells in turn recruit even more leukocytes. Although it is likely that increased levels of SDF-1 due to DPP4 inhibition show positive effects in diabetic patients (for detailed information please see S Vedantham et al., Cells 2018, 7, 181; doi:10.3390/cells7100181), it is not clear what might happen in diabetic patients infected with SARS-CoV-2. Although SDF-1 mediated increased levels of leukocytes may show positive effects in the early phase of COVID-19 in terms of combating viral infection, the question is what happens when mast cells get recruited and activated (what is very likely in SARS-CoV-2 infected patients) and in turn recruit even more leukocytes? Does this result in adverse effects such as sustained severe inflammation as observed in COVID-19 patients? In that context, the role of mast cells in COVID-19 should be discussed in detail and the authors should comment on the positive or may be even negative effects of mast cells in COVID-19.
Authors: We now discuss the potential dual role of mast cells in SARS-CoV-2 infection, as it may add controversy in the debate about the use of DPP4 inhibitors by means of protection against a severe illness (see lines 129-144).
Moreover, I think it would be very helpful for the reader to show a sketch explaining the mechanisms potentially resulting in positive or negative effects (if existing) of DPP4 inhibitors in diabetic patients with SARS-CoV-2 infection.
Authors: We have now added an illustration concerning the potential benefit and harms of DPP4 blockage in COVID-19 (see Figure 1, pag.4).
I personally would be very happy of DPP4 inhibitors would show beneficial effects in COVID-19 patients, but for me this is not really obvious after reading this short article. However, this review article offers an excellent opportunity to discuss the potential effects of DPP4 inhibitors in detail and either support or contradict the use this drug class in COVID-19.
Together, I think it needs to be very critically addressed whether DPP4 inhibitors may show beneficial or harmful effects in diabetic (and may be also non-diabetic) individuals infected with SARS-CoV-2. The Abstract and the conclusion should be modified accordingly.
Authors: According to the reviewer’s suggestion, we have now expanded section 2 (pages 2-4) for a better clarification of the potential benefits of DPP4 inhibitors in diabetic patients exposed to SARS-CoV-2, addressing controversies and/or potential harmful effects (lines 108-111; 142-144). Abstract and conclusions have been also modified (lines 22; 313-314; 316-319).
GLP1 receptor agonists: The paper would again profit from a figure illustration the effect of GLP1 receptor agonists on upregulation of ACE2 along with the potential positive and negative effects in corona infected patients.
Authors: We have now added an illustration concerning the potential benefits and harms of GLP1 receptor agonists in COVID-19 (see Figure 2, pag.7).
Table 1: Please supply references for statements on efficacy and safety issues for each drug mentioned.
Authors: We have now added a “reference” column in Table 1.
Reviewer 2 Report
This mini review article by Mirabelli summarized the recent discovery on the relations between some antidiabetic drugs and COVID-19 treatments. The authors reviewed the reported mechanisms on how the antidiabetic drugs would affect the cytokine release syndrome and the consequent effects on the treatment of COVID-19-infected patients with diabetes using the antidiabetic drugs. It was concluded that DPP4 inhibitors may be employed to prevent/reduce COVID-19-related manifestations, while SGLT2 inhibitors and GLP2 receptor agonists should be suspended for patients. To best of my knowledge, this is the first review article on this topic with nice organization. It is ready to be published.
Author Response
This mini review article by Mirabelli summarized the recent discovery on the relations between some antidiabetic drugs and COVID-19 treatments. The authors reviewed the reported mechanisms on how the antidiabetic drugs would affect the cytokine release syndrome and the consequent effects on the treatment of COVID-19-infected patients with diabetes using the antidiabetic drugs. It was concluded that DPP4 inhibitors may be employed to prevent/reduce COVID-19-related manifestations, while SGLT2 inhibitors and GLP2 receptor agonists should be suspended for patients. To best of my knowledge, this is the first review article on this topic with nice organization. It is ready to be published.
Authors: We thank the reviewer for the nice comments!
Reviewer 3 Report
COVID-19 is a disease caused by a novel coronavirus infection. I has caused huge suffering people. How to manage diabetic patients with COVID-19 is becoming important issue. This communication provides some information.
Please add the following information: 1. What are the differences between treatment on diabetes with COVID-19 infection and diabetes without COVID-19 infection; 2. What are the factors of COVID-19 to influence the outcomes of treatment?
Author Response
Authors: We thank the reviewer for the careful reading of the manuscript.
COVID-19 is a disease caused by a novel coronavirus infection. I has caused huge suffering people. How to manage diabetic patients with COVID-19 is becoming important issue. This communication provides some information.
Please add the following information: 1. What are the differences between treatment on diabetes with COVID-19 infection and diabetes without COVID-19 infection; 2. What are the factors of COVID-19 to influence the outcomes of treatment?
Authors: We have now added a new section “5. Optimizing glycemic control” (page 7) concerning this issue. However, the purpose of our communication was not to speculate on the potential mechanisms for glycemic derangement observed in critically-ill COVID-19 patients, as insulin therapy would be the only feasible therapeutic option to regain glycemic control in these cases, irrespective of the previous metabolic state. We aimed to briefly discuss whether DPP4 inhibitors, SGLT2 inhibitors, and GLP1 receptor agonists would retain the potential to exacerbate or attenuate the clinical presentation of COVID-19 in patients with pre-existing diabetes, as little is currently known about this topic.
Round 2
Reviewer 1 Report
The manuscript significantly improved and an lot of important information is supplied. However, two little question are still not answered:
Severe COVID-19 is described to be associated with damage of small vessels. Is there any information available, whether this small vessels are capillaries, venules, or arterioles? If yes, please specify the vessels. If not, please inform the reader that there is no information available about the kind of vessels, which are affected. I think this is an important point also concerning the potential positive or negative function of mast cells, which are located in the perivascular space of arterioles. And also concerning the potential positive or negative function of leukocytes as leukocyte diapedesis occurs in post-capillary venules.
Another question which is still open is whether SARS-CoV-2 activates or deactivates DPP4. Is there any information available? Please inform the reader.
Author Response
Reviewer: The manuscript significantly improved and an lot of important information is supplied. However, two little question are still not answered:
Severe COVID-19 is described to be associated with damage of small vessels. Is there any information available, whether this small vessels are capillaries, venules, or arterioles? If yes, please specify the vessels. If not, please inform the reader that there is no information available about the kind of vessels, which are affected. I think this is an important point also concerning the potential positive or negative function of mast cells, which are located in the perivascular space of arterioles. And also concerning the potential positive or negative function of leukocytes as leukocyte diapedesis occurs in post-capillary venules.
Authors: As suggested by the reviewer, we now refer to pathological evidences of pulmonary capillary damage and microthrombosis in severe COVID-19 illness (see lines 39-44).
Reviewer: Another question which is still open is whether SARS-CoV-2 activates or deactivates DPP4. Is there any information available? Please inform the reader.
Authors: To date, there is no information available concerning this issue, as discussed in lines 148-150.